# Feedback Attention for Cell Image Segmentation

**Abstract.** In this paper, we address cell image segmentation task by Feedback Attention mechanism like feedback processing. Unlike conventional neural network models of feedforward processing, we focused on the feedback processing in the human brain and assumed that the network learns like a human by connecting feature maps from deep layers to shallow layers. We propose some Feedback Attentions which imitate human brain and feeds back the feature maps of output layer to close layer to the input. U-Net with Feedback Attention showed better result than the conventional methods using only feedforward processing.

**Keywords:** Cell image, Semantic Segmentation, Attention Mechanism, Feedback Mechanism

## 1 Introduction

Deep neural networks has achieved state-of-the-art performance in image classification [19], segmentation [22], detection [25], and tracking [3]. Since the advent of AlexNet [19], several Convolutional Neural Network (CNN) [20] has been proposed such as VGG [28], ResNet [13], Deeplabv3+ [5], Faster R-CNN [25], and Siamese FC [3]. These networks are feedfoward processing. Neural network is mathematical model of neurons [34] that imitate the structure of the human brain. Human brain performs not only feedfoward processing from shallow layers to deep layers of neurons, but also feedback prooocessing from deep layers to shallow layers. However, conventional neural networks consist of only feedfoward prooocessing from shallow layers to deep layers, and do not use feedback processing to connect from deep layers to shallow layers. Therefore, in this paper, we propose some Feedback Attention methods using position attention mechanism and feedback process.

Semantic segmentation assigns class labels to all pixels in an image. The study of this task can be applied to various fields such as automatic driving [4, 6], cartography [11, 23] and cell biology [10, 17, 26]. In particular, cell image segmentation requires better results in order to ensure that cell biologists can perform many experiments at the same time.

In addition, overall time and cost savings are expected to be achieved by automated processing without human involvement to reduce human error. Manual segmentation by human experts is slow to process and burdensome, and there is a significant demand for algorithms that can do the segmentation quickly and accurately without human. However, cell image segmentation is a difficult task

because the number of supervised images is smaller and there is not regularity compared to the other datasets such as automatic driving. A large number of supervised images requires expert labeling which takes a lot of effort, cost and time. Therefore, it is necessary to enhance the segmentation ability for pixel-level recognition with small number of training images.

Most of the semantic segmentation approaches are based on Fully Convolutional Network (FCN) [22]. FCN is composed of some convolutional layers and some pooling layers, which does not require some fully connected layers. Convolutional layer and pooling layer reproduce the workings of neurons in the visual cortex. These are proposed in Neocognitron [9] which is the predecessor of CNN [20]. Convolutional layer which is called S-cell extracts local features of the input. Pooling layer which is called C-cell compresses the information to enable downsampling to obtain position invariance. Thus, by repeating the feature extraction by convolutional layer and the local position invariance by pooling layer, robust pattern recognition is possible because it can react only to the difference of shape without much influence of misalignment and size change of the input pattern. Only the difference between CNN [20] and Neocognitron [9] is the optimization method, and the basic elements of both are same structure.

We focused on the relationship between the feature map close to the input and output of the semantic segmentation, and considered that it is possible to extract effective features by using between the same size and number of channels in the feature maps close to the input and output. In this paper, we create an attention map based on the relationship between these different feature maps, and a new attention mechanism is used to generate segmentation results. We can put long-range dependent spatial information from the output into the feature map of the input. The attention mechanism is feedback into the feature map of the input to create a model that can be reconsidered in based on the output.

In experiments, we evaluate the proposed method on two kinds of cell image datasets  [10]. We confirmed that the proposed method gave higher accuracy than conventional method. We evaluate our method by some ablation studies and show the effectiveness of our method.

This paper is organized as follows. In section 2, we describe related works. The details of proposed method are explained in section 3. In section 4, we evaluate our proposed method on segmentation of cell images. Finally, we describe conclusion and future works in section 5.

## 2    Related works

### 2.1    Semantic Segmentation

FCNs [22] based methods have achieved significant results for semantic segmentation. The original FCN [22] used stride convolutions and pooling to gradually downsize the feature map, and finally created high-dimensional feature map with low-resolution. This feature map has semantic information but fine information such as fine objects and correct location are lost. Thus, if the upsampling is used

at the final layer, the accuracy is not sufficient now. Therefore, encoder-decoder structure is usually used in semantic segmentation to obtain a final feature map with high-resolution. It consists of encoder network that extracts features from input image using convolutional layers, pooling layers, and batch normalization layers, and decoder network that classifies the extracted feature map by upsampling, convolutional layers, and batch normalization layers. Decoder restores the low-resolution semantic feature map extracted by encoder and middle-level features to the original image to compensate for the lost spatial information, and obtains a feature map with high resolution semantic information.

SegNet [2] is a typical network of encoder-decoder structures. Encoder uses 13 layers of VGG16 [28], and decoder receives some indexes selected by max pooling of encoder. In this way, decoder complements the positional information when upsampling and accelerates the calculation by unpooling, which requires no training.

Another famous encoder-decoder structural model is U-net [26]. The most important characteristic of U-Net [26] is skip connection between encoder and decoder. The feature map with the spatial infomation of encoder is connected to the restored feature map of the decoder. This complements the high-resolution information and improves the resolution so that labels can be assigned more accurately to each pixel. In addition, deconvolution is used for up-sampling in decoder.

## 2.2    Attention Mechanism

Attention mechanism is an application of the human attention mechanism to machine learning. It has been used in computer vision and natural language processing. In the field of image recognition, important parts or channels are emphasized.

Residual Attention Network [31] introduced a stack network structure composed of multiple attention components, and attention residual learning applied residual learning [13] to the attention mechanism. Squeeze-and-Excitation Network (SENet) [15] introduced an attention mechanism that adaptively emphasizes important channels in feature maps. Accuracy booster blocks [29] and efficient channel attention module [32] made further improvements by changing the fully-connected layer in SENet [15]. Attention Branch Network [8] is Class Activation Mapping (CAM) [38] based structure to build visual attention maps for image classification. Transformer [30] performed language translation only with the attention mechanism. There are Self-Attention that uses the same tensor, and Source-Target-Attention that uses two different tensors. Several networks have been proposed that use Self-Attention to learn the similarity between pixels in feature maps [7, 16, 24, 33, 36].

## 2.3   Feedback Mechanism using Recurrent Neural Networks

Feedback is a fundamental mechanism of the human perceptual system and is expected to develop in the computer vision in the future. There have been several approaches to feedback using recurrent neural networks (RNNs) [1, 12, 35].

Feedback Network [35] uses convLSTM to acquire hidden states with high-level information and provide feedback with the input image. However, this is intended to solve the image classification task and is not directly applicable to the segmentation task.

RU-Net [1] consists of a U-Net [26] and a recurrent neural network, where each convolutional layer is replaced by recurrent convolutional layer [21]. The accumulation of feature information at each scale by the recurrent convolutional layer gives better results than the standard convolutional layer. However, this is not strictly feedback but the deepening of network.

Feedback U-Net [27] is the segmentation method using convLSTM. The probability for segmentation at final layer is used as the input image for segmentation at the second round, while the first feature map is used as the hidden state for the second segmentation to provide feedback.

Since RNNs is a neural network that contains loop connections, it can be easily used for feedback mechanisms. However, the problem with RNNs is that the amount of operations increases drastically and a lot of memory is consumed, which makes processing difficult and often results in the phenomenon that information is not transmitted. Thus, we applied RNNs-free feedback mechanism to U-Net, and excellent performance is shown by the feedback attention mechanism on the segmentation task.

## 3   Proposed Method

This section describes the details of the proposed method. Section 3.1 outlines the network of the proposed method. In section 3.2, we describe the details of the proposed attention mechanism.

### 3.1   Network Structure Details

The proposed method is based on U-Net [26], which is used as a standard in medical and cell images. Figure. 1 shows the detail network structure of our proposed method using U-net [26]. We design to do segmentation twice using U-Net [26] in order to use the feature maps in input and output. Since the proposed method uses the feature maps of input and output, we use the model twice with shared weights. First, we perform segmentation by U-Net to obtain high-resolution important feature maps at the final layer. Then, we connect to Feedback Attention to a feature map that is close to the input with the same size and number of channels as this feature map. In this case, we use the input feature map that was processed two times by convolution.

The reason is that a feature map convolved twice can extract more advanced features than a feature map convolved once. The details of Feedback Attention

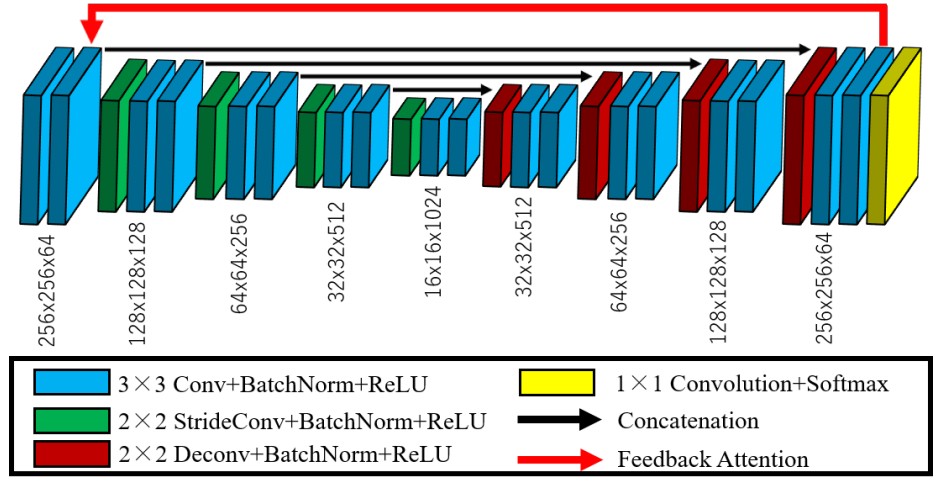

**Fig. 1.** Network structure of the proposed method using Feedback Attention

is explained in section 3.2. By applying Attention between the feature maps of input and output, we can obtain an input that takes the output into account, as feedback control. In training, U-Net is updated by using only the gradients at the second round using feedback attention. In addition, the loss function is trained using Softmax cross-entropy loss.

## 3.2    Feedback Attention

We propose two kinds of Feedback Attentions to aggregate feature maps with the shape of $C \times H \times W$. Figure. 2 (a) shows the Source-Target-Attention method that directly aggregates similar features between the feature maps of input and output. Figure. 2 (b) shows the self-attention method that performs self-attention for output feature map and finally adds it to the feature map of input. Both Feedback Attentions are explained in the following subsections.

**Feedback Attention using Source-Target-Attention** We use Source-Target-Attention to aggregate the correlation between feature maps based on the relationship between input and output. Since the feature map in the final layer close to the output contains all the information for judging, it can be fed back using attention and effectively extract features again from the shallow input layer. We elaborate the process to aggregate each feature map.

As shown in Figure. 2 (a), we feed the feature maps of input or output into $1 \times 1$ convolutions and batch normalization to generate two new feature maps **Query** and **Key**, respectively, we are inspired by Self-Attention GAN (SAGAN) [36] to reduce the channel number to $C/8$ for memory efficiency. Then, we reshape them to $C/8 \times (H \times W)$. After we perform a matrix multiplication

(a) Source-Target-Attention

(b) Self-Attention

**Fig. 2.** Feedback Attention

between the transpose of **Query** and **Key**, and we use a softmax function to calculate the attention map. Attention map in vector form is as follows.

$$w_{ij} = \frac{1}{Z_i} \exp(Query_i^T \; Key_j), \qquad (1)$$

where $w_{ij}$ measures the $i^{th}$ **Query**'s impact on $j^{th}$ **Key**. $Z_i$ is the sum of similarity scores as

$$Z_i = \sum_{j=1}^{H \times W} \exp(Query_i^T \; Key_j), \qquad (2)$$

where $H \times W$ is the total number of pixels in **Query**. By increasing the correlation between two locations, we can create an attention map that takes into account output's feature map.

On the other hand, we feed the feature map of output into $1 \times 1$ convolution and batch normalization to generate a new feature map **Value** and reshape it to $C/2 \times (H \times W)$. Then, we perform a matrix multiplication between attention map and the transpose of **Value** and reshape the result to $C \times H \times W$. Finally, we multiply it by a scale parameter $\alpha$ and perform a element-wise sum operation with the input feature map to obtain the final output as follows.

$$A_i = \alpha \sum_{j=1}^{H \times W} (w_{ij} \ Value_j^T)^T + F_i, \tag{3}$$

where $\alpha$ is initialized as 0 and gradually learns to assign more weight [36]. $A_i$ indicates the feedbacked output and $F_i$ indicates the feature map of the input. By adding $\alpha \sum_{j=1}^{H \times W} (w_{ij} \ Value_j^T)^T$ to the feature map close to input, we can get the feature map considering feature map of output. The new feature map $A_i$ is fed into the network again, and we obtain the segmentation result.

From Equation. (3), it can be inferred that the output $A_i$ is the weighted sum of all positions in output and the feature map of input. Therefore, the segmentation accuracy is improved by transmitting the information of the output to the input.

**Feedback Attention using Self-Attention** In Source-Target-Attention, the feature map between input and output is aggregated. Thus, the relationship between each feature map can be emphasized. However, the feature map of the input may not extract enough information and therefore may result in poorly relational coordination. We construct Feedback Attention using Self-Attention that aggregates only the feature map of output.

The structure is shown in Figure. 2 (b). We feed the feature maps of output into $1 \times 1$ convolution and batch normalization to generate new feature maps **Query**, **Key** and **Value**. This is similar to Source-Target-Attention. We also reshape **Query** and **Key** to $C/8 \times (H \times W)$. Then, we perform a matrix multiplication between the transpose of **Query** and **Key**, and use a softmax function to calculate the attention map. Attention map in vector form is as follows.

$$w_{pq} = \frac{\exp(Query_p^T \ Key_q)}{\sum_{q=1}^{H \times W} \exp(Query_p^T \ Key_q)}, \tag{4}$$

where $w_{pq}$ measures the $p^{th}$ **Query**'s impact on $q^{th}$ **Key**.

We reshape **Value** to $C/2 \times (H \times W)$. Then, we perform a matrix multiplication between attention map and the transpose of **Value** and reshape the result to $C \times H \times W$. Finally, we multiply it by a scale parameter $\beta$ and perform a element-wise sum operation with the feature maps of input to obtain the final output as follows.

$$A_p = \beta \sum_{q=1}^{H \times W} (w_{pq} \ Value_q^T)^T + F_p, \tag{5}$$

where $\beta$ is initialized as 0 and gradually learns to assign more weight [36]. $A_p$ indicates the output, $F_p$ indicates the feature map of input. New feature map $A_p$ is fed into the network again, and we obtain the segmentation result.

Unlike Equation. (3), Equation. (5) calculates the similarity using only the information of output. In addition, consistency can be improved because information can be selectively passed to the input by the scale parameter.

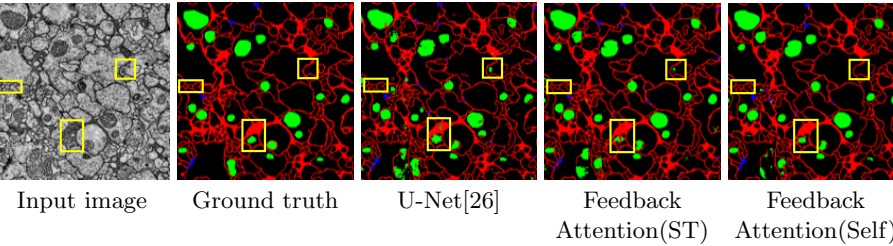

|              |              |          | Feedback         | Feedback           |
| Input image  | Ground truth | U-Net[26]| Attention(ST)    | Attention(Self)    |

**Fig. 3.** Examples of segmentation results on ssTEM dataset. ST indicates Source-Target-Attention, Self indicates Self-Attention.

**Table 1.** Segmentation accuracy (IoU and mIoU) on ssTEM Dataset. ST indicates Source-Target-Attention, Self indicates Self-Attention.

| Method | Membrane | Mitochondria | Synapse | Cytoplasm | Mean IoU% |
|---|---|---|---|---|---|
| U-Net[26] | 74.24 | 71.01 | 43.08 | 92.03 | 70.09 |
| RU-Net[1] | 75.81 | 74.39 | 43.26 | 92.25 | 71.43 |
| Feedback U-Net[27] | 76.44 | 75.20 | 42.30 | 92.43 | 71.59 |
| Feedback Attention(ST) | **76.65** | **78.27** | **43.32** | **92.64** | **72.72** |
| Feedback Attention(Self) | 76.94 | 79.52 | 45.29 | 92.80 | 73.64 |

## 4   Experiments

This section shows evaluation results by the proposed method. We explain the datasets used in experiments in section 4.1. Experimental results are shown in section 4.2. Finally, section 4.3 describes Ablation studies to demonstrate the effectiveness of the proposed method.

### 4.1   Dataset

In experiments, we evaluated all methods 15 times with 5-fold cross-validation using three kinds of initial values on the Drosophila cell image data set [10]. We use Intersection over Union (IoU) as evaluation measure. Average IoU of 15 times evaluations is used as final measure.

This dataset shows neural tissue from a Drosophila larva ventral nerve cord and was acquired using serial section Transmission Electron Microscopy at HHMI Janelia Research Campus [10]. This dataset is called ssTEM dataset. There are 20 images of $1024 \times 1024$ pixels and ground truth. In this experiment, semantic segmentation is performed for four classes; membrane, mitochondria, synapses and cytoplasm. We augmented 20 images to 320 images by cropping 16 regions of $256 \times 256$ pixels without overlap from an image. We divided those images into 192 training, 48 validation and 80 test images.

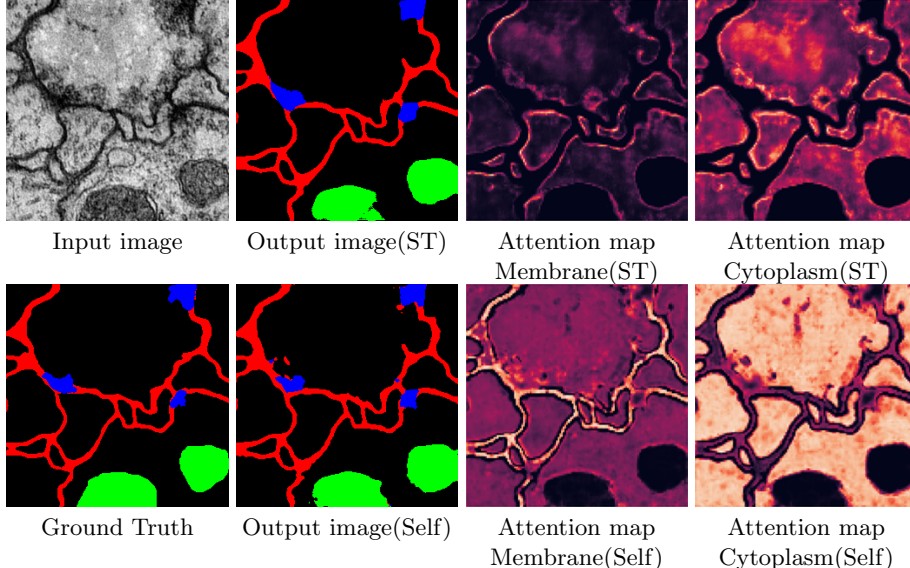

|  |  |
|---|---|
| Input image | Output image(ST) |
| Ground Truth | Output image(Self) |

| Attention map Membrane(ST) | Attention map Cytoplasm(ST) |
| Attention map Membrane(Self) | Attention map Cytoplasm(Self) |

**Fig. 4.** Visualization results of Attention Map on ssTEM dataset. ST indicates Source-Target-Attention, Self indicates Self-Attention.

### 4.2    Experimental Results

Table. 1 shows the accuracy on ssTEM dataset, and Figure. 3 shows the segmentation results. Bold red letters in the Table represent the best IoU and black bold letters represent the second best IoU. Table. 1 shows that our proposed Feedback Attention improved the accuracy of all classes compared to conventional U-Net [26]. We also evaluated two feedback methods using RNNs; RU-Net [1] with recurrent convolution applied to U-Net and Feedback U-Net [27] with feedback segmentation applied to U-Net. The result shows that the proposed method gave high accuracy in all classes. In addition, we can see that Self-Attention, which calculates the similarity in the output, is more accurate than Source-Target-Attention which calculates the similarity from the relationship between the input and the output. This indicates that the feature map of the input does not extract enough features and therefore the similarity representation between the input and the output does not work well.

From the yellow frames in Figure. 3, our method using Feedback Attention can identify mitochondria that were detected excessively by conventional methods. In the conventional methods, the cell membranes were interrupted, but in our proposed method, we confirm that the cell membranes are segmented in such a way that they are cleanly connected. Experimetal results show that the cell membrane and the mitochondria have been successfully identified even in places where it is difficult to detect by conventional methods.

We visualize some attention maps in Figure. 4 to understand our two kinds of Feedback Attentions. White indicates similarity and black indicates dissimilar-

**Table 2.** Comparison of different feedback connections.

| Method | Membrane | Mitochondria | Synapse | Cytoplasm | Mean IoU% |
|---|---|---|---|---|---|
| Add | 75.56 | 77.36 | 41.84 | 92.46 | 71.81 |
| 1*1 Conv | 75.22 | **78.39** | **43.46** | 92.49 | 72.39 |
| SE-Net[15] | 75.89 | 77.31 | 42.92 | 92.49 | 72.15 |
| Light Attention[14] | 76.20 | 78.27 | 43.18 | 92.57 | 72.56 |
| Feedback Attention(ST) | **76.65** | 78.27 | 43.32 | **92.64** | **72.72** |
| Feedback Attention(Self) | **76.94** | **79.52** | **45.29** | **92.80** | **73.64** |

ity. We find that Self-Attention maps has many similar pixels but Source-Target-Attention maps has fewer pixels. This is because Source-Target-Attention uses the feature maps of input and output, and the feature map near input is different from that of output, so the number of similar pixels are smaller than Self-Attention map. However, the menbranes and cytoplasm have different values in the attention map. This means that they are emphasized as different objects. On the other hand, Self-Attention generates attention maps from only the feature map of output. Therefore, as shown in the figure 4, when the cell membrane and cytoplasm are selected, they are highlighted as similar pixels.

### 4.3    Ablation Studies

We performed two ablation studies to show the effectiveness of the proposed method. One ablation study evaluated the different connection methods. The other ablation study confirmed the effectivenss of connection location from the output to the input.

**Comparison of difference feedback connection** The effectiveness of the other feedback connection methods from the output to the input was experimentally confirmed. We compare four methods. We compare two methods that do not use the attention mechanism. The firrst one is that we simply add the feature map in the output to the input. The second one is that we feed the feature map in the output to $1 \times 1$ convolution and then add it to the feature map in the input. Both methods use scale parameter as our propose method.

In addition, we compare two methods using attention mechanism. The first one is that we apply SE-Net [15], which suppresses and emphasizes the feature map between channels, to the output feature map, and add it to the input feature map. The second one is that we apply Light Attention [14], which suppresses and emphasizes the important locations and channels in feature map by $3 \times 3$ convolutional processing, to the output feature map and adding it to the input feature map.

From Table. 2, we can see that the above four methods improve the accuracy from U-Net [26] because the feedback mechanism is effective. However,

**Table 3.** Comparison between different connection locations.

| Method | Membrane | Mitochondria | Synapse | Cytoplasm | Mean IoU% |
|---|---|---|---|---|---|
| Feedback Attention using Source-Target-Attention | | | | | |
| One conv | 76.54 | 77.39 | 43.06 | 91.96 | 72.24 |
| Two conv(Ours) | 76.65 | 78.27 | 43.32 | 92.64 | 72.72 |
| Feedback Attention using Self-Attention | | | | | |
| One conv | **76.69** | **78.73** | **45.23** | **92.66** | **73.33** |
| Two conv(Ours) | **76.94** | **79.52** | **45.29** | **92.80** | **73.64** |

**Table 4.** Comparison before and after Feedback Attention.

| Method | Membrane | Mitochondria | Synapse | Cytoplasm | Mean IoU% |
|---|---|---|---|---|---|
| Feedback Attention using Source-Target-Attention | | | | | |
| First output | 76.07 | 76.76 | 41.28 | 92.39 | 71.62 |
| Second output(Ours) | **76.65** | **78.27** | **43.32** | **92.64** | **72.72** |
| Feedback Attention using Self-Attention | | | | | |
| First output | 75.49 | 74.29 | 41.57 | 92.03 | 70.84 |
| Second output(Ours) | **76.94** | **79.52** | **45.29** | **92.80** | **73.64** |

our proposed method is more accurate than those four methods. This shows that our proposed method Feedback Attention can use the output's information effectively in the input.

**Comparison between different connection locations** We experimentally evaluated the location of the input feature map which is the destination of feedback. Since the size of feature map should be the same as final layer, the candidates are only two layers close to input. The first one is the feature map closest to the input which is obtained by only one convolution process. The other one is the feature map obtained after convolution is performed two times. We compared the two feature map locations that we use Feedback Attention.

Table. 3 shows that the Feedback Attention to the feature map after two convolution process is better for both Source-Target-Attention and Self-Attention. This indicates that only one convolution process does not extract good features than two convolution processes.

**Comparison before and after Feedback Attention** When We use Feedback Attention, the output of network is feedback to input as attention. Thus, we get the ouputs twice. Although we use the output using Feedback Attention at the second round is used as final result, we compare the results of the outputs at the first and second rounds to show the effectiveness of Feedback Attention. From Table. 4, the output using Feedback Attention as the second round is better than that at first round. This demonstrates that the accuracy was improved through the feedback mechanism.

## 5   Conclusions

In this paper, we have proposed two Feedback Attention for cell image segmentation. Feedback Attention allows us to take advantage of the feature map information of the output and improve the accuracy of the segmentation, and segmentation accuracy is improved in comparison with conventional feedforward network, RU-Net [1] which uses local feedback and Feedback U-Net [27] which uses global feedback. Ablation studies show that Feedback Attention can obtain accurate segmentation results by choosing the location and attention mechanism that conveys the output information.

In the future, we aim to develop a top-down attention mechanism that directly utilizes ground truth, such as self-distillation [37]. Feedback networks are also categorized as a kind of top-down networks, because the representation of feature extraction will be expanded if the ground truth can be used for direct learning in the middle layer as well. In addition, Reformer [18] using Locality Sensitive Hashing has been proposed in recent years. Since Transformer-based Attention uses a lot of memory, Reformer will work well in our Feedback Attention. These are subjects for future works.

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
