# OpenReview forum: "Feedback Attention for Cell Image Segmentation"
_thecvf.com/ECCV/2020/Workshop/BIC — BIC 2020 Oral_

### Official Review · AnonReviewer2 · 2020-07-30
**Thoroughly-investigated addition of feedback attention to U-Net segmentation**

**Rating:** 8
**Confidence:** 3

**Review:**

### Quality
This work approaches the addition of a feedback attention mechanism into a U-Net based segmentation method with commendable rigour. While the authors only demonstrate their method on one dataset, they do so in a methodical manner whereby the carefully validate what sort of feedback works best, and also offer explanations as to why this is the case. The authors also validate which layers of the network work most effectively for their chosen network architecture, and that the improvement in segmentation performance is indeed due to the addition of feedback attention rather than just from running data through the U-Net twice.

The figures are generally good quality. The attention maps in Figure 4 are particularly striking and are a good demonstration of the difference between the two proposed feedback attention methods introduced here.

I would have been very interested to see the authors look closer at the attention maps and performance for the mitochondria and synapse classes, as these classes display the most striking segmentation accuracy improvement over the other methods compared. It would also have been interesting for the authors to have discussed cross-applicability of this approach to other microscopy modalities, for example digital pathology data.

### Clarity
It took me a few reads of the paper in order to sufficiently understand the method, and there are quite a lot of grammatical errors and typos throughout.

I personally found that the Query/Key/Value terminology was quite clunky and detracted from my ability to understand section 3.2. For example, the phrase ‘the $i$th Query’s impact’ (line 255) threw me for a while, as I was unclear which dimension (C, H, W) $i$ was indexing. This may well be my own unfamiliarity with the field, but being slightly more explicit in the explanation of the indices would have helped me understand much quicker.

It also felt like there was too much unnecessary repetition between the explanations of the Source-Target-Attention and Self-Attention methods. Again, if I understand correctly, the Source-Target-Attention and Self-Attention methods are identical except for the ‘Query’ being identical to the ‘Key’ in the latter case? If so, this could have been displayed in a more compact mathematical way.

### Originality
This work appears to be the first instance in which attention is integrated into a U-Net via a feedback mechanism. I am not an expert in this field, but a cursory literature search retrieved a [paper discussing the use of attention in medical image segmentation via attention gates](https://arxiv.org/abs/1804.03999) – perhaps this should have been mentioned as ‘Related Work’

### Significance
High-quality semantic segmentation is an undoubtedly significant and impactful challenge for microscopy. The technical merit of this work has been made clear, but I think that the authors could have expanded on the significance of the improved performance in the field of cell imaging. For example, the conclusion does not mention the application at all, and more dwells on further technical adaptations, which feels a little short-sighted. For that reason I also have concerns regarding deployment of this technique and ensuring that biologists are actual able to reap the benefits for their research.

### Pros
* Excellent technical rigour demonstrated in investigating novel method through ablation studies
* Significant increase in segmentation performance achieved with this method

### Cons
* The sections describing the generation of attention maps were quite difficult to follow and understand
* While the performance is very good, the authors do not discuss a route by which this method can actually be used for the benefit of the application demonstrated.

**Reviews Visibility:**

I agree that my anonymized review is made publicly visible, if the submission is accepted.

---

### Official Review · AnonReviewer3 · 2020-07-31
**Simple and effective U-Net feedback extension**

**Rating:** 8
**Confidence:** 5

**Review:**

Summary
-------

This paper presents a feedback mechanism for U-Nets, which re-uses the output
of the U-Net for a second round of processing. For the incorporation of the
output into an earlier feature map of the U-Net, the authors propose two
attention mechanisms (source-target-attention and self-attention). On an
electron microscopy dataset of neural tissue, experiments demonstrate
consistent improvements of the self-attention version of the proposed method
over several baselines, including a vanilla U-Net and a feedback U-Net. In
further ablation studies, the authors investigate the two proposed attention
mechanisms, the choice of the injection point of the output, and the usefulness
of the second round of processing.

Quality and Clarity
-------------------

The technical description of the method is very clear and the provided figures
helpful.

Originality
-----------

Reusing the output of a U-Net for a second round of processing is not an
entirely new idea (as acknowledged by the authors in the Related Work section).
The main contribution here is therefore the addition of an attention mechanism
and the re-use of the same weights for the second round of processing.

Significance
------------

The experimental evaluation is thorough (albeit on only a single
dataset) and addresses key questions about the proposed method. Qualitative
results (especially the attention maps generated by the two proposed
mechanisms) are helpful to understand the contributions of the method.

Pros
----

* clear presentation
* elegant architecture
* convincing results
* thorough analysis of method components in ablation study

Cons
----

* evaluated on only one dataset

Minor Comments
--------------

* line 44: "cell image segmentation is a difficult task because [...] there is not regularity compared to other datasets such as automatic driving" I would personally argue that the opposite is true
* line 420: "menbranes" -> "membranes"
* line 437: "firrst" -> "first"
* line 487: "We" -> "we"
* no "." after "Equation", "Table", or "Figure"
* line 71: "we evaluate the proposed method on two kinds of cell image datasets" results are only presented on one dataset


**Reviews Visibility:**

I agree that my anonymized review is made publicly visible, if the submission is accepted.

---

### Decision · Program_Chairs · 2020-07-31

Accept (Oral)